# Antimicrobial and Diagnostic Stewardship of the Novel β-Lactam/β-Lactamase Inhibitors for Infections Due to Carbapenem-Resistant Enterobacterales Species and *Pseudomonas aeruginosa*

**DOI:** 10.3390/antibiotics13030285

**Published:** 2024-03-21

**Authors:** Stefanos Ferous, Cleo Anastassopoulou, Vassiliki Pitiriga, Georgia Vrioni, Athanasios Tsakris

**Affiliations:** Department of Microbiology, Medical School, National and Kapodistrian University of Athens, 11527 Athens, Greece; sferous@med.uoa.gr (S.F.); cleoa@med.uoa.gr (C.A.); vpitiriga@med.uoa.gr (V.P.); gvrioni@med.uoa.gr (G.V.)

**Keywords:** carbapenem resistance, carbapenemase-producing pathogens, metallo-β-lactamase, KPC, OXA-48-like, detection method, algorithm, antimicrobial stewardship

## Abstract

Carbapenem-resistant Gram-negative bacterial infections are a major public health threat due to the limited therapeutic options available. The introduction of the new β-lactam/β-lactamase inhibitors (BL/BLIs) has, however, altered the treatment options for such pathogens. Thus, four new BL/BLI combinations—namely, ceftazidime/avibactam, meropenem/vaborbactam, imipenem/relebactam, and ceftolozane/tazobactam—have been approved for infections attributed to carbapenem-resistant Enterobacterales species and *Pseudomonas aeruginosa*. Nevertheless, although these antimicrobials are increasingly being used in place of other drugs such as polymyxins, their optimal clinical use is still challenging. Furthermore, there is evidence that resistance to these agents might be increasing, so urgent measures should be taken to ensure their continued effectiveness. Therefore, clinical laboratories play an important role in the judicious use of these new antimicrobial combinations by detecting and characterizing carbapenem resistance, resolving the presence and type of carbapenemase production, and accurately determining the minimum inhibitor concentrations (MICs) for BL/BLIs. These three targets must be met to ensure optimal BL/BLIs use and prevent unnecessary exposure that could lead to the development of resistance. At the same time, laboratories must ensure that results are interpreted in a timely manner to avoid delays in appropriate treatment that might be detrimental to patient safety. Thus, we herein present an overview of the indications and current applications of the new antimicrobial combinations and explore the diagnostic limitations regarding both carbapenem resistance detection and the interpretation of MIC results. Moreover, we suggest the use of alternative narrower-spectrum antibiotics based on susceptibility testing and present data regarding the effect of synergies between BL/BLIs and other antimicrobials. Finally, in order to address the absence of a standardized approach to using the novel BL/BLIs, we propose a diagnostic and therapeutic algorithm, which can be modified based on local epidemiological criteria. This framework could also be expanded to incorporate other new antimicrobials, such as cefiderocol, or currently unavailable BL/BLIs such as aztreonam/avibactam and cefepime/taniborbactam.

## 1. Introduction

Antimicrobial resistance represents a major threat to public health and healthcare systems worldwide. Approximately 4.9 million deaths were attributed to antimicrobial resistance in 2019 [1], while its economic cost might reach one trillion USD by 2050 [2]. Among antimicrobial-resistant pathogens, carbapenem-resistant Gram-negative bacteria (CR-GNB) pose a significant threat to public health due to their resistance to most β-lactams in addition to other antimicrobial agents [3].

However, four novel β-lactam/β-lactamase inhibitors (BL/BLIs)—namely, ceftazidime/avibactam (CZA), meropenem/vaborbactam (MEV), imipenem/relebactam (IMR), and ceftolozane/tazobactam (C/T)—have been approved for the treatment of various CR-GNB infections, enhancing our antibiotic arsenal. Although these antimicrobial combinations have been in clinical use for approximately eight years, there is evidence that bacterial resistance to these agents might be increasing [4,5], thus demonstrating the importance of constant antimicrobial and diagnostic stewardship. Furthermore, the inherent heterogeneity and variable sensitivity and specificity of the available diagnostic tests used to characterize carbapenem resistance complicates the use of these antibiotics.

Herein, we present an overview of carbapenemases and novel BL/BLIs and subsequently discuss the role of BL/BLIs alone or in combination with other antimicrobials in the treatment of CR-GNB infections. We also discuss methods to approach discordant minimum inhibitory concentrations (MICs) and phenotypic results during susceptibility testing, and, finally, we propose a diagnostic and therapeutic algorithm based on regional epidemiological data, which will allow for the rapid and streamlined identification and subsequent treatment of CR-GNB isolates.

## 2. Overview of Resistance Mechanisms to Carbapenems

Carbapenems are a family of β-lactam antibiotics that includes meropenem, imipenem, doripenem, and ertapenem. They have a broad spectrum of activity against numerous antimicrobial-resistant Gram-positive and Gram-negative pathogens. Carbapenem resistance was first identified in 1997 [6] and has spread globally ever since [7]. The primary resistance mechanism is the production of carbapenemases, which are β-lactamase enzymes capable of hydrolyzing most β-lactam antibiotics, including carbapenems [8]. Other resistance mechanisms include multidrug efflux pumps and a reduced expression of porins. Although infections by carbapenemase-producing isolates are typically hospital-acquired, the spread of carbapenemase genes to zoonotic pathogens such as *Salmonella enterica* raises concerns for the permeation of carbapenem resistance across the food chain, affecting individuals without prior exposure to broad-spectrum antibiotics [9].

Carbapenemases belong to Ambler classes A, B, and D and are capable of hydrolyzing aminopenicillins, ureidopenicillins, narrow- and broad-spectrum cephalosporins, cephamycins, and carbapenems [10]. The most important and common class A carbapenemase is the *Klebsiella pneumoniae* carbapenemase (KPC), which has spread amongst numerous Enterobacterales species. Examples of class D carbapenemases are the OXA-derived carbapenemases, commonly found in *Acinetobacter baumannii.* Certain OXA-carbapenemases, such as the OXA-48-like carbapenemases, have spread amongst Enterobacterales and have become an important mechanism of carbapenem resistance among such isolates in several countries, including Turkey [11,12]. Metallo-β-lactamases (MBLs) are class B carbapenemases and include (i) the New Delhi MBLs (NDMs), which are common in India and Pakistan but have also spread globally [13,14], (ii) the Verona integron-encoded MBL (VIM) types, and (iii) various imipenemase (IMP) types. Initially, MBL production was commonly found in *Pseudomonas aeruginosa* but has also spread to several Enterobacterales species [15]. MBL producers are particularly difficult to treat due to their frequent co-expression of other resistance genes against numerous antimicrobial classes [16,17]. It is noteworthy that aztreonam (AZT) is a β-lactamic antibiotic that is resistant to the enzymatic activity of MBLs but not to that of class A and D carbapenemases [13,16]. Most carbapenemases are plasmid-mediated, which allows for the rapid dissemination of resistance among different bacterial strains [12,18].

It is also important to note that not all carbapenemases have the same affinity for carbapenems. For example, OXA-48-like carbapenemases have a low affinity for carbapenems but are usually co-expressed with numerous other mechanisms, such as efflux pumps and porin deletions which synergistically increase carbapenem resistance [18]. Also, not all carbapenemases of the same group have the same hydrolyzing capability for all carbapenems. For example, clinical isolates of *K. pneumoniae* that produce mutated IMP enzymes have been identified, which confer resistance to meropenem but not imipenem [19].

## 3. The New β-Lactam and β-Lactamase Inhibitor Combinations: Indications and Resistance Mechanisms

The spread of carbapenem-resistant Enterobacterales and *P. aeruginosa* is a global concern due to the few treatment options available. Four novel BL/BLIs have been approved for the treatment of infections caused by these strains, and an overview of all these combinations is available in Table 1. Current guidelines do not propose the use of these combinations in the treatment of carbapenem-resistant *A. baumannii* [20,21].

### 3.1. Meropenem/Vaborbactam

The combination of MEV was approved in 2017 by the FDA to treat complicated urinary tract infections (cUTIs) only; however, it has been used off-label for other infections caused by non-CR-GNB [22]. The EMA approved the combination to treat cUTIs, complicated intra-abdominal infections (cIAIs), as well as hospital-acquired pneumonia (HAP) cases [23]. Vaborbactam is a broad-spectrum β-lactamase inhibitor that inhibits serine carbapenemases and Ambler class A extended-spectrum β-lactamase (ESBL) enzymes. It also inhibits AmpC enzymes, but it does not inhibit MBLs or class D ESBLs and carbapenemases such as OXA-derived ESBLs and OXA-derived carbapenemases [24]. MEV has been proven effective in treating infections caused by KPC-producing Enterobacterales [25]. However, it is not considered effective against carbapenem-resistant *P. aeruginosa* [26].

OmpK35 and OmpK36 porins are important for vaborbactam function, and, frequently, the downregulation or mutation of these porins is associated with the development of resistance. OmpK36 mutations appear to have a greater effect on vaborbactam efficacy, as demonstrated by Lomovskaya et al., with gene deletions and amino acid duplications resulting in reduced vaborbactam efficacy [24]. A loss of transcriptional factors associated with porin gene expression has also been shown to reduce vaborbactam efficacy [27]. It is of note that isolates with porin mutations frequently co-express other resistance mechanisms including the increased expression of KPCs [28,29]. Multidrug efflux pumps may also contribute to MEV resistance when co-expressed with other resistance mechanisms [24].

### 3.2. Ceftazidime/Avibactam

CZA was approved by the FDA in 2015 for the treatment of cIAI and cUTI and, in 2018, for the treatment of HAP and ventilator-associated pneumonia (VAP) [30]. The EMA also approved the combination for the same indications in 2016. Avibactam has an identical spectrum to vaborbactam, in addition to having the ability of inhibiting certain D class carbapenemases, such as OXA-48-like enzymes [31], and, thus, can be used to treat infections caused by KPC- and OXA-48-producing Enterobacterales [21]. CZA can be considered a treatment option against carbapenem-resistant *P. aeruginosa*, but, in general, it should not be used without documented MIC susceptibility testing due to its unpredictable efficacy [32,33].

No benefit was noted when CZA was used in combination therapies [34], and guidelines do not recommend combination therapies in infections caused by CR-GNB who are susceptible to CZA or MEV. However, CZA can be used in combination therapies with AZT to treat infections caused by MBL-producing isolates. AZT is inherently resistant to MBLs, and avibactam readily inhibits the action of AmpC, ESBL, and KPC β-lactamases that could hydrolyze AZT [35]. CZA + AZT could be used for the treatment of MBL-producing *P. aeruginosa* [36]. Interestingly, although avibactam does not inhibit MBL action, in vitro studies demonstrate that the exposure of MBL-producing bacteria to avibactam increases bacterial clearance by increasing the susceptibility of isolates to complement neutrophil and cathelicidin function [37].

CZA resistance is primarily associated with KPC mutations or the increased expression of KPC enzymes [38,39]. CZA appears to be more susceptible to specific KPC variants, such as KPC-3, with KPC-3 having an increased affinity to ceftazidime [40,41]. Moreover, KPC-3 mutants that confer resistance to CZA are, surprisingly, frequently susceptible to meropenem [42,43]. These mutations, which primarily affect the Ω-loop, increase the enzyme’s affinity to ceftazidime while simultaneously reducing its affinity to avibactam [43]. Such mutations, however, confer an ESBL-like phenotype to the enzyme, thus restoring carbapenem efficacy. These mutants are frequently associated with false-negative phenotypic test results [44]. Isolated clinical reports have demonstrated the efficacy of meropenem combinations in treating CZA-resistant isolates due to these KPC mutations [45]. However, Shields et al. demonstrated that serial passages of such isolates through media containing sublethal meropenem concentrations resulted in the selection of carbapenem-resistant isolates that retain their CZA resistance profile [42]. Vaborbactam appears to be effective against certain KPC mutants that result in CZA resistance [46]. Finally, porin mutations responsible for MEV resistance are frequently associated with CZA resistance [47,48].

### 3.3. Imipenem/Relebactam

IMR was approved by the FDA in 2019 for the treatment of cIAI and cUTIs and, in 2020, for HAP and VAP [49]. The EMA approved the combination for HAP, VAP, and bacteremia caused by susceptible strains [50]. Moreover, the EMA approved the use of this combination in any infection caused by aerobic Gram-negative bacteria that are susceptible and for which no other treatment option is available. Relebactam readily inhibits class A carbapenemases; however, it exhibits variable activity against class D carbapenemases [51]. It has no effect on MBLs. IMR is the only combination that has demonstrated efficacy against *P. aeruginosa* infections [52], possibly by enhancing the stability and activity of imipenem [53].

Resistance to IMR remains the least studied of all these combinations. Epidemiological data indicate that resistance is primarily associated with the production of carbapenemases not inhibited by relebactam [54], with certain studies demonstrating a role for ompK35 and ompK36 mutations [55].

### 3.4. Ceftolozane/Tazobactam

Ceftolozane is a novel fifth generation cephalosporin, which was approved by the FDA in 2014 for the treatment of cIAIs and UTIs and, in 2019, for the treatment of HAP and VAP [56]. It has also been approved by the EMA with the same indications [57]. Ceftolozane’s action is compromised when exposed to ESBLs and carbapenemases, and the subsequent addition of tazobactam restores ceftolozane’s action against certain ESBL-producing bacteria [58]. The addition of tazobactam does not increase the antipseudomonal activity of ceftolozane [59]. Ceftolozane is, nevertheless, resistant to the action of AmpC β-lactamases and not affected by multidrug resistance efflux pump systems, such as the MexAB-OprM system, or by the deletion of porins, which are commonly observed mechanism of carbapenem resistance in *P. aeruginosa* [60]. The combination is, therefore, used primarily in the treatment of multidrug-resistant *P. aeruginosa* infections and for the treatment of non-carbapenemase-producing CR-GNB. It has already been demonstrated to be non-inferior to carbapenems in the setting of HAP and VAP caused by Gram-negative organisms [61], and one retrospective study has demonstrated the efficacy of C/T over polymyxin- or aminoglycoside-based combinations in *P. aeruginosa* infections [62]. C/T resistance is less frequent than IMR resistance in *P. aeruginosa* [63,64].

The most well-characterized mechanism is the mutation of an AmpC-type cephalosporinase, PDC. The overexpression or mutation of this enzyme, both of which would increase the rate of ceftolozane hydrolysis, results in resistance amongst *P. aeruginosa* isolates [65]. In addition, mutated Ambler C class β-lactamases and the inactivation of AmpC-negative regulators have also been implicated [66]. Moreover, C/T can be ineffective against isolates that produce carbapenemases or ESBLs that are not inhibited by tazobactam [67].

## 4. Future Directions in Antimicrobial and Diagnostic Stewardship of Novel β-Lactam/β-Lactamase Inhibitors

Clinical laboratories play an important role in the judicious use of novel BL/BLIs by detecting and characterizing carbapenem resistance, resolving the presence and type of carbapenemase production, and accurately determining the MICs for BL/BLIs. These three targets must be met to ensure optimal BL/BLIs use and prevent unnecessary exposure that can lead to the development of resistance.

Phenotypic tests are the methods primarily deployed by diagnostic laboratories to detect and characterize carbapenemase production. Examples of such tests include combination disc testing (CDT), colorimetric tests such as the Carba-NP, lateral flow assays such as the NG-Test Carba-5, and the carbapenem inactivation method (CIM). There are no recommendations or guidelines on the optimal way of using these detection methods, and each laboratory uses different methods based on their ease of use, cost, and availability. A review of the advantages and limitations of the most common phenotypic detection tests is provided in Table 2. Genotypic testing methods are more sensitive and can detect the presence of efflux pumps or porin deletions that can contribute to carbapenem resistance [68]. However, they are associated with increased costs and require specialized equipment. EUCAST does not recommend the direct detection of carbapenemase genes among carbapenemase-producing Enterobacterales [69,70].

Current laboratory practices leave much to be desired both in terms of the efficacy and accuracy of current methods. Moreover, the rapid emergence of resistance to the novel BL/BLIs dictates that antimicrobial use must be targeted, efficient, and rational. To address these issues, we hereby discuss the role of antimicrobial synergy for the treatment of CR-GNB, identify methods to accurately combine MIC and phenotypic test results, and, finally, propose a diagnostic and therapeutic framework based on objective epidemiological data to streamline CR-GNB identification and treatment.

## 5. Antibiotic Synergy and Treatment of Carbapenem-Resistant Enterobacterales

CZA has been proven to be an important addition to our therapeutic armamentarium for treating various infections, such as those caused by KPC- or OXA-48-like-producing CR-GNB. However, resistance and treatment failures appear when CZA is used as a single antimicrobial agent [39,44]. Nevertheless, CZA has been found to act synergistically with carbapenems against carbapenem-resistant KPC producers. In that respect, a recent study has demonstrated excellent synergistic effects with a fractional inhibitory concentration index (FICI) < 0.5 when CZA was combined with meropenem or AZT even in isolates that were resistant to CZA, meropenem, and AZT [93]. Similarly, Gaibani et al. showed that the combination of CZA with imipenem or meropenem resulted in a synergistic effect (FICI < 0.5) in all *K. pneumoniae* isolates, including two which exhibited resistance to CZA due to mutations in the *bla*_KPC-3_ gene [94]. In both studies, the combination resulted in the restoration of carbapenem efficacy. Another study demonstrated a synergistic effect of MEV with CZA and imipenem even in initially CZA- and MEV-resistant strains [95]. Zhang et al. demonstrated the efficacy of CZA in combination with imipenem in the treatment of XDR-*P. aeruginosa* isolates that do not produce MBL but possess porin and AmpC mutations and multidrug efflux pumps and overexpress AmpC enzymes. Although counterintuitive, the efficacy of this combination may be explained by the reduction in the AmpC levels by ceftazidime, which acts as a suicide molecule, and by the inhibition of the AmpC enzymes by avibactam, in addition to the known reduced affinity of imipenem to efflux pumps [96].

Finally, the use of novel BL/BLIs synergies has been studied in the treatment of double carbapenemase producers. Double carbapenemase producers are frequently associated with high-carbapenem MICs and an extensive resistance profile, reducing the available antimicrobial agents [97]. The treatment of such isolates is further complicated by the inability of many phenotypic testing methods to accurately detect and characterize double carbapenemase production (Table 1). In theory, the combination of CZA + AZT is a rational choice for strains producing multiple carbapenemases, due to the proven efficacy of the combination against MBL producers and the known efficacy of CZA against OXA enzymes. Romina et al. have demonstrated the synergistic effect of CZA + AZT in carbapenemase-producing isolates, including those with a double production of KPC plus NDM [98], and Jayol et al. have demonstrated the efficacy of this combination against strains that co-produce NDM and OXA enzymes [99]. Additionally, it has been demonstrated that both CZA + AZT and MEV + AZT combinations can be used for the treatment of double carbapenemase producers, with MEV + AZT possibly being more effective [100].

Synergies have also been reported with other antibiotics such as fosfomycin, polymyxins, and aminoglycosides [101]. It has been shown that CZA + colistin is effective in producing more than two-log reductions in all CZA-resistant *P. aeruginosa* isolates [102]. A case report has also demonstrated the synergistic effect of MEV with fosfomycin in the treatment of a patient with a severe *K. pneumoniae* infection. The patient did not respond to CZA therapy and subsequently relapsed after MEV was administered. The relapsing strain was resistant to MEV, but the patient was successfully treated with the addition of fosfomycin [103].

Therefore, numerous in vitro data suggest that antimicrobial synergies could expand the spectrum of novel BL/BLIs to initially resistant isolates, thus providing new treatment options for MDR and XDR pathogens. Moreover, these synergies could reduce the emergence of resistance amongst susceptible isolates [93].

## 6. Interpreting Carbapenem and Novel BL/BLI MIC Results in CR-GNB

EUCAST recommends further screening for carbapenemase production in all Enterobacterales isolates with a meropenem MIC > 0.125 mg/L, which corresponds to a disk-diffusion zone diameter of <28 mm [69,70]. These screening MIC and zone diameter cutoffs proposed by EUCAST are different from the clinical breakpoints, which characterize an isolate as either susceptible or resistant. This reflects the fact that certain carbapenemase producers can be classified as being susceptible based on clinical cutoff criteria due to the low affinity of the produced carbapenemase for the selected carbapenem in susceptibility testing or even due to the low carbapenemase production rates [69,104]. In fact, a recent study has demonstrated that approximately 50% of NDM producers were susceptible to meropenem, and approximately 60% of NDM producers were susceptible to imipenem [105]. However, pursuing further testing in many isolates that are within the clinical susceptibility range but harbor carbapenem resistance genes can result in the overuse of last-line antibiotics including the novel BL/BLIs [106].

These conflicting results present a therapeutic dilemma. Should MIC be the sole criterion for carbapenem use or should phenotypic and/or genotypic detection of a carbapenemase, regardless of the MIC values, automatically preclude the use of a carbapenem? A 2015 study demonstrated that the meropenem that is used in combination therapies can be used in the treatment of infections caused by Gram-negative bacteria with reduced susceptibility to meropenem (i.e., MIC ≤ 8 mg/L) in high-risk patients [107]. This is reflected in the 2022 ESCMID guidelines, according to which high-dose extended-infusion meropenem therapy in combination with other in vitro susceptible antibiotics is proposed as an alternative, albeit with a low certainty of evidence, to novel BL/BLI combinations, provided that the MIC remains ≤8 mg/L [21]. This dilemma is not novel and is frequently encountered in infections caused by ESBL bacteria. Ever since the MERINO trial, carbapenems have been the mainstay for the treatment of ESBL infections [108]. However, the use of piperacillin/tazobactam for the treatment of UTIs and biliary tract infections caused by susceptible ESBL *Escherichia* coli and *K. pneumoniae* strains has been shown to be safe and effective [109], with one retrospective study concluding that amoxicillin/clavulanic acid and piperacillin/tazobactam were non-inferior to carbapenem when used in bacteremia caused by susceptible strains of *E. coli* [110]. Therefore, one could conclude that carbapenem combination therapies could be used provided that meropenem’s MIC is ≤8 mg/L. Colistin could further augment carbapenem action, possibly reducing treatment failures and the emergence of resistance and minimizing novel BL/BLI use [111].

Another important parameter that can affect the efficacy of carbapenems in infections caused by carbapenemase-producing bacteria is the inoculum effect, which refers to the increase in MIC as the bacterial inoculum increases. In vitro studies have shown that the inoculum effect can have a profound effect on the efficacy of carbapenems in carbapenemase-producing Enterobacterales, which depends on the underlying carbapenem resistance mechanism and the type of carbapenemase produced. Adler et al. have shown that non-carbapenemase resistance mechanisms (e.g., porin mutations) are rarely associated with an inoculum effect [112], whereas Golikova et al. have demonstrated that OXA-48 producers exhibit a less-pronounced inoculum effect compared to KPC and NDM producers [113]. Therefore, infections caused by OXA producers or non-carbapenemase producers, initially susceptible to meropenem, are less likely to be associated with clinical failures, since an initial MIC result within the susceptible range will not be severely affected by a higher inoculum. Conversely, high-inoculum infections should not be treated with carbapenems regardless of the initial MIC.

Similar to carbapenems, discordant MICs and phenotypic results have been described for the novel BL/BLI combinations. A recent study demonstrated that 57% of MBL producers tested susceptible to MEV [114]. These results could be caused by laboratory misidentification, a phenomenon more common with commercially available identification methods such as automated systems or Etest strips. CZA Etests have been shown to result in very major errors (i.e., false susceptibility) in up to 6% of isolates and, thus, might not fulfill the ISO performance standards [115]; thus, the results obtained from CZA Etest strips should be cautiously interpreted [116]. Clinicians and clinical microbiologists need to be aware of the limitations of current commercially available MIC determination techniques for novel BL/BLIs when interpreting the MICs for these agents, especially when discordant results are present. These results need to be verified via reference standard methods (broth microdilution or disc diffusion) and additional phenotypic or genotypic testing. A true discordant result can be attributed either to a low production of the detected carbapenemase or a low affinity of the carbapenemase for the tested combination. Regardless, few data are available addressing this phenomenon, in a similar fashion to the use of carbapenems for carbapenemase producers mentioned previously. Therefore, we suggest that phenotypic results be the deciding factor for treatment.

## 7. Carbapenemase Detection—Rational Testing and Treatment Options Based on Epidemiological Data

The absence of a systematic approach to the detection and characterization of carbapenem resistance might be attributed to the numerous pitfalls surrounding the available diagnostic tests [117,118] (see Table 2). Diagnostic uncertainties remain, especially in relation to the detection of OXA-48-like carbapenemase producers, the identification of carbapenemase production in *P. aeruginosa* isolates, and the proper characterization of bacteria with complex carbapenem resistance mechanisms (e.g., production of numerous carbapenemases of different types and/or co-expression of non-carbapenemase-related resistance mechanisms). These uncertainties translate to ambiguous treatment decisions.

Another important factor that needs to be accounted for is time. The early initiation of an appropriate antimicrobial therapy is associated with improved survival in patients with bloodstream infections, and the accurate characterization of the underlying resistance mechanisms in carbapenem-resistant organisms is time-consuming, considering the extra time needed for some phenotypic tests such as CDT. The entire process, starting from the moment in which a blood culture is drawn to final carbapenemase detection and MIC reporting, for novel BL/BLI combinations could extend over more than 72 h [119] (Figure 1). Although this time gap can be bridged using molecular methods, their cost and need for expensive equipment preclude their widespread use.

Therefore, there is a need for a clear diagnostic framework that could accurately detect most carbapenemase producers whilst simultaneously providing clear guidelines on which antibiotic regimen should be prescribed pending confirmatory testing and MIC results. In addition, streamlined processes that allow for the rapid identification of resistant isolates would allow for the notification of clinicians and administrators in order to deploy infection control measures such as nurse and patient cohorting. These measures are necessary in order to curb the horizontal transmission of plasmid-mediated carbapenemase genes [120]. Here, we propose a diagnostic and treatment algorithm based on key epidemiological data that could be altered based on local and national epidemiology and provide the example of Greece, a country with a high burden of CR-GNB infections. The algorithm is centered around an initial rapid diagnostic test that will allow for the initiation of treatment based on local epidemiology on the day when carbapenem resistance is detected. Subsequent confirmatory testing can be pursued as dictated by the epidemiology via CDT or CIM or even genotypic testing in selected cases.

Epidemiological data such as what are the most common mechanisms of carbapenem resistance among Enterobacterales and *P. aeruginosa* strains should be addressed to pursue epidemiologically appropriate diagnoses and treatments. For example, in Greece, the most common carbapenem-resistant microorganism is *K. pneumoniae*, with most strains being KPC producers. However, non-carbapenemase mechanisms are also common, and certain studies have demonstrated that double carbapenemase production (e.g., KPC + VIM) can be encountered in up to 25% of isolates [121,122,123]. OXA-type carbapenemase production is uncommon in non-*Acinetobacter* cases, with rare isolations of *K. pneumoniae* strains co-expressing OXA-48-like enzymes in addition to other carbapenemases [122]. Epidemiological data regarding the resistance phenotypes of other carbapenem-resistant Gram-negative bacteria such as *E. coli* are missing, but no double carbapenemase production has been reported. Epi-net data indicate that carbapenem-resistant *E. coli* strains are presumed to be carbapenemase producers [124]. Studies have shown that 35% of *P. aeruginosa isolates* are carbapenem-resistant, with most of them expressing numerous resistance mechanisms. As far as carbapenemases are concerned, most carbapenemase-producing *P. aeruginosa* strains produce MBLs [125]. Based on these epidemiological data, diagnostic and therapeutic algorithms are proposed in Figure 2 and Figure 3. These algorithms are proposed for three different microorganisms: *K. pneumoniae,* non-*K. pneumoniae* Enterobacterales (most commonly *E. coli*) species, and *P. aeruginosa*.

*K. pneumoniae* isolates can produce KPC, MBL, or both and might even possess other non-carbapenemase-related resistance mechanisms. A colorimetric test such as the Blue Carba Test (BCT) or the Carba NP cannot distinguish carbapenemase types and, therefore, has limited use for the initial screening of carbapenem-resistant *K. pneumoniae*. NG-Test Carba-5 testing is a reasonable choice due to its ability to detect most carbapenemases as well as double carbapenemase producers. If the rapid test is positive for a double carbapenemase (such as KPC + VIM or KPC + NDM), the treatment options include CZA + AZT or tigecycline- or polymyxin-based therapies. A subsequent CDT, primed for detecting double carbapenemase producers, is a suitable confirmatory test [71,72]. Genotypic testing is also not unreasonable, considering the relatively unknown sensitivity of most phenotypic testing in accurately detecting double producers. If the initial rapid test is positive for one carbapenemase type, the diagnostic algorithm continues with CDT, with the aim of confirming the absence of double carbapenemase producers. The initial treatment should be tailored to the initial rapid results: if the result is KPC-positive, treatment is recommended with MEV; if it is MBL-positive, CZA + AZT should be administered; and, if the result is OXA-48-positive, CZA should be used. If CDT confirms the initial rapid testing result, no further diagnostic action is needed. If the CDT and NG-Test Carba-5 results are discordant, treatment should be changed to CZA + AZT, and genotypic testing is considered mandatory in this case.

A negative NG-Test Carba-5 result has two possible interpretations: either the isolate is carbapenem-resistant due to non-carbapenemase-related mechanisms or the production of a rare carbapenemase type (e.g., GES carbapenemase), or, alternatively, the result is a false negative. In all cases, genotypic testing is recommended to determine the presence of non-carbapenemase-mediated resistance. Although CZA + AZT could be considered for treatment, polymyxin- or tigecycline-based therapies are recommended pending confirmatory genotypic and MIC results. In all isolated *K. pneumoniae* isolates, synergy testing with CZA + AZT is recommended due to the high rate of MBL production. Laboratories should consider adding additional antimicrobial synergy testing based on the novel BL/BLI combinations and colistin with carbapenems, aminoglycosides, and fosfomycin for all double carbapenemase producers, as indicated by NG-Test Carba 5, and for all patients with severe diseases. This is based on studies that, although demonstrating no differences in terms of a reduction in mortality between combination and monotherapies, have cited possible indication biases as a cause for the lack of benefit seen from combination therapies in addition to the delays associated with initiating CZA therapies [126].

Determining carbapenem resistance in non-*K. pneumoniae* Enterobacterales, such as carbapenem-resistant *E. coli,* is less-challenging considering the more predictable phenotype (Figure 3). A simple KPC or MBL Etest strip [127] could suffice as an initial and cheap screening tool, while NG-Test Carba 5 is a suitable alternative. If the initial screening test is positive for KPC production, MEV should be used. If an MBL is detected, then CZA + AZT should be provided. In the latter case, CZA + AZT synergy should be tested in vitro. A confirmatory CDT could also be suggested, since the detection of MBL by means of MBL Etests, especially when EDTA is used as the MBL inhibitor, has been associated with false-positive results [128]. If however, initial testing with either Etest strips or NG-Test Carba-5 are negative, genotypic testing is highly recommend, and treatment with polymyxin- or tigecycline-based combinations are needed.

Finally, *P. aeruginosa* isolates should initially be approached with a simple colorimetric test (Figure 3). The high specificity of the test means that a negative result excludes carbapenemase production, and, thus, C/T can be administered pending MIC results. Genotypic results are also recommended, but not expected, to provide valuable data that would alter treatment decisions. An NG-Test Carba-5 might also be a suitable alternative.If the test is positive and a carbapenemase is detected (most likely an MBL), the initial treatment should be polymyxin-based, which, while more toxic, might be safer, considering the unpredictable MICs for CZA, IMR, and CZA + AZT [17]. The determination of CZA and IMR MICs and CZA + AZT synergy is considered necessary for carbapenemase-producing *P. aeruginosa* prior to their use. Moreover, a further characterization of the underlying mechanisms of resistance using molecular methods is highly recommended.

It is of note that different countries can develop various diagnostic approaches based on these epidemiological data. For example, in the USA, KPC production predominates, with double carbapenemase producers being a rare exception [121,129]. Therefore, treatment can be initiated with a positive NG-Test Carba-5, a KPC and MBL Etest strip, or a modified colorimetric test. Subsequent confirmatory testing, if deemed necessary, should be conducted with EDTA-based CIM (eCIM) and modified-CIM (mCIM), since there is no need for double carbapenemase testing.

The diagnostic and therapeutic frameworks proposed can easily be altered and augmented to add new antimicrobial combinations, especially against MBL-producing isolates. The combination of AZT/avibactam has already been demonstrated to be non-inferior to meropenem, without or with colistin, in the treatment of severe Gram-negative infections. Data related to the efficacy of this combination against MBL producers are awaited [130]. In addition, the combination of cefepime with a novel β-lactamase inhibitor, taniborbactam, the first inhibitor which is active against all β-lactamase classes, has been shown to be superior to meropenem in a recent clinical trial featuring patients with cUTIs [131]. More trials are awaited in order to compare the efficacy of these two combinations against CZA + AZT or colistin-based regimens in the treatment of infections caused by MBL-producing Enterobacterales and *P. aeruginosa.* In addition, cefiderocol, a siderophore cephalosporin resistant to most β-lactamases, including carbapenemases, is increasingly being used for the treatment of multidrug-resistant Gram-negative infections, including carbapenem-resistant Enterobacterales and *P. aeruginosa* [132]. Although clinical experience with this agent is limited, it has shown promise as a last-line therapy against numerous carbapenem-resistant isolates, including carbapenem-resistant *P. aerugionsa* and *A. baumanni* [17,133]. The versatility of streamlined algorithmic processes allows for treatment flexibility based on newer guidelines.

There are, however, certain limitations with our proposition, and the drafting of the algorithm proposed in this paper is based on certain assumptions: (1) rare carbapenemase types such as GES-type carbapenemases are indeed rare among Enterobacterales and *P. aeruginosa* strains, and, as such, the false-negative rate of the initial NG-Test Carba-5 screening is low; (2) the resistance phenotype of other non-*K. pneumoniae* carbapenem-resistant Enterobacterales (mainly *E. coli*) is associated primarily with class A carbapenemases, with no double carbapenemase production; and (3) the patient at hand has not been recently exposed to novel BL/BLIs. In such cases, the probability of mutations either in the carbapenemase itself, in the produced porins, or in efflux pumps is relatively high, and, thus, phenotypic diagnostic tests may have significant disadvantages.

## 8. Conclusions

Novel BL/BLI combinations have provided new possibilities for the treatment of infections caused by certain CR-GNB. However, given the rising resistance to these agents and the difficulties surrounding current carbapenemase identification methods, adjustments in our diagnostic and therapeutic approaches are needed. Here, we presented in vitro evidence for the use of the novel BL/BLIs as single agents as well as in synergies with other antimicrobials for treating carbapenem-resistant isolates, a practice which might increase BL/BLIs efficacy and reduce resistance rates. Moreover, we discussed the importance of properly interpreting MIC and phenotypic testing results, especially in isolates with conflicting results, and proposed means to identify isolates that might benefit from narrower-spectrum therapies, such as carbapenems, thus sparing the use of the novel BL/BLIs. Finally, to address the absence of a standardized approach to detect carbapenemase production, we proposed a diagnostic algorithm, based on epidemiological data, that will streamline diagnostic pathways and ensure sensible broad antimicrobial coverage, with proper de-escalation when necessary.

## Figures and Tables

**Figure 1 antibiotics-13-00285-f001:**
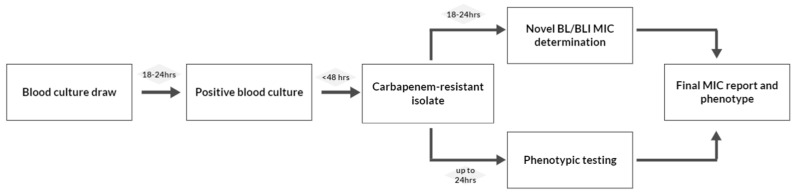
Current laboratory process in detecting and characterizing carbapenemase producers. BL/BLI: β-lactam/β-lactamase inhibitor.

**Figure 2 antibiotics-13-00285-f002:**
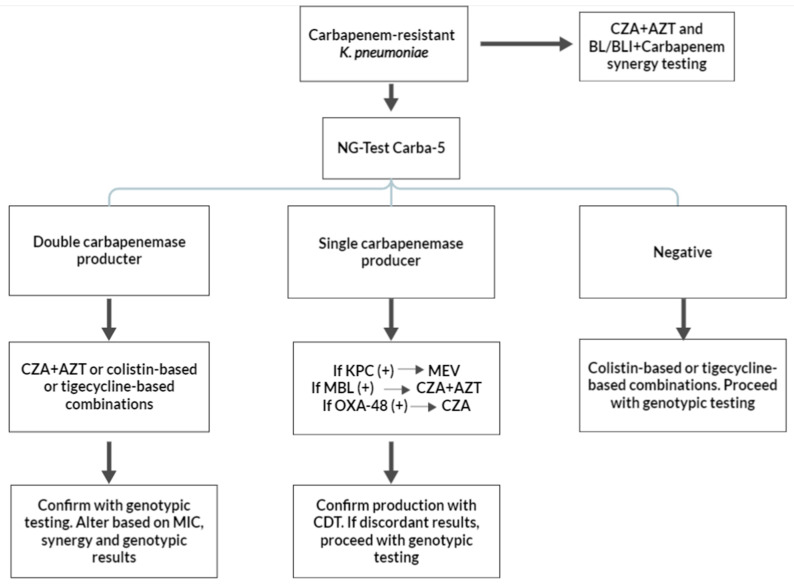
Suggested diagnostic and treatment algorithms of carbapenem-resistant *Klebsiella pneumoniae* based on carbapenemase production. KPC: *Klebsiella pneumoniae* carbapenemase; MBL: metallo-β-lactamase; MEV: meropenem/vaborbactam; CZA: ceftazidime/avibactam; AZT: aztreonam; CDT: combination disc testing; and OXA-48: OXA-48 oxacillinase.

**Figure 3 antibiotics-13-00285-f003:**
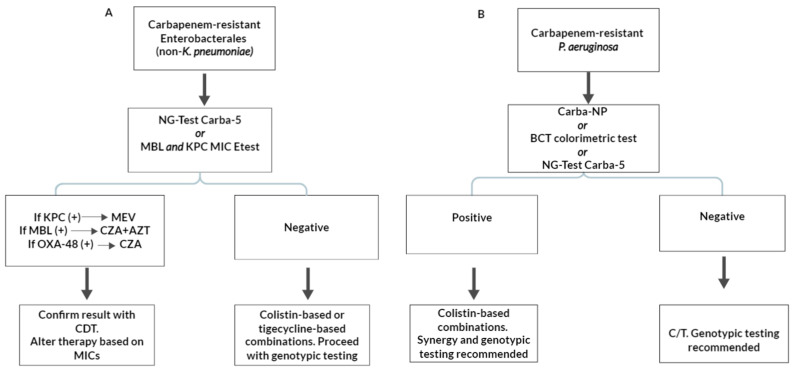
Suggested diagnostic and treatment algorithms for carbapenem-resistant Enterobacterales (non-*Klebsiella pneumoniae*) (**A**) and *Pseudomonas aeruginosa* (**B**) based on carbapenemase production. KPC: *Klebsiella pneumoniae* carbapenemase; MBL: metallo-β-lactamase; MEV: meropenem/vaborbactam; CZA: ceftazidime/avibactam; AZT: aztreonam; CDT: combination disc testing; BCT: Blue Carba Test; and C/T: ceftolozane/tazobactam.

**Table 1 antibiotics-13-00285-t001:** Novel BL/BLI combinations and their susceptibility based on carbapenemase type.

Antibiotic Combination	KPC	MBL	OXA	CRPA
Ceftazidime/avibactam	+	−	+	+/−
Ceftazidime/avibactam + Aztreonam	+	+	+	+/−
Meropenem/vaborbactam	+	−	−	−
Imipenem-cilastatin/relebactam	+	−	+/−	+/−
Ceftolozane/tazobactam	−	−	−	+/−

+: susceptible; −: not susceptible; and +/−: conditionally susceptible (see text). KPC: *Klebsiella pneumoniae* carbapenemase; MBL: metallo-β-lactamase; OXA: oxacillinase carbapenemases; and CRPA: carbapenem-resistant *P. aeruginosa.*

**Table 2 antibiotics-13-00285-t002:** Advantages and limitations of major phenotypic carbapenemase detection methods.

Detection Method	Advantages	Limitations	Notes	References
CDT	- Differentiates between carbapenemase types. - Relatively easy to conduct and interpret. - Low-cost. - Not affected by rare carbapenemases.	- Limited OXA detection. Use of temocillin as a surrogate marker for OXA production. - Limited sensitivity in detecting double carbapenemase producers. - Time-consuming. - Sensitivities vary by manufacturer.	Modifications that increase sensitivity to OXA; double producers exist, but they are not commercially available or standardized.	[70,71,72,73,74,75]
Colorimetric methods	- Rapid carbapenemase detection (<2 h). - High sensitivity and specificity. - Relatively low cost. - Simple to conduct. - Not affected by rare carbapenemases.	- Limited sensitivity in OXA producers. - Possibly lower sensitivity in detecting carbapenemases in *A. baumannii* isolates. - Most do not differentiate between carbapenemase types.	Modifications exist that can detect different carbapenemase types and even resistance to meropemen–vaborbactam. The Blue Carba Test (BCT) appears to be more sensitive than the CarbaNP test in detecting OXA producers.	[76,77,78,79,80,81]
NG-Test CARBA-5	- Rapid detection of five most common carbapenemases (KPC, VIM, IMP, NDM, and OXA-48). - High sensitivity and simple to conduct. - Can differentiate between different carbapenemases. - Identifies double producers.	- False positives for NDM reported. - Possibly reduced efficacy in detecting carbapenemase-producing *P. aeruginosa*. - Cost ranging from USD 10 to USD 20 per sample. - Rare carbapenemase types not detected.		[82,83,84]
CIM	- Simple and low-cost. - Not affected by rare carbapenemases. - Numerous modifications can increase the detection rates of specific carbapenemases.	- Cannot detect double producers. - Many modifications are not commercially available. - Results read after overnight incubation and might extend to approximately 20 h.	Guidelines propose the use of mCIM and eCIM. eCIM detects MBL producers by utilizing EDTA.	[70,85,86,87,88]
MALDI-TOF	- Can detect most carbapenemases in addition to non-carbapenemase-related resistance mechanisms. - Can detect resistance to novel BL/BLI combinations.	- High cost of installation. - Requires trained personnel to operate and interpret the results.		[89,90,91,92]

CDT: combination disc testing; CIM: carbapenem inactivation method; KPC: *Klebsiella pneumoniae* carbapenemase; VIM: Verona integron-mediated carbapenemase; NDM: New-Delhi metallo-β-lactamase; OXA-48: oxacillinase-48 carbapenemase; mCIM: modified CIM test; and eCIM: EDTA-based CIM test.

## Data Availability

All the data of this study are included in this article.

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
