# Peer review of "Antimicrobial and Diagnostic Stewardship of the Novel β-Lactam/β-Lactamase Inhibitors for Infections Due to Carbapenem-Resistant Enterobacterales Species and Pseudomonas aeruginosa"

_antibiotics, 2024, doi:10.3390/antibiotics13030285_

Round 1

Reviewer 1 Report

Comments and Suggestions for Authors

The work is interesting and provides a comprehensive overview of the challenges and advancements in the treatment of carbapenem-resistant gram-negative bacterial infections using new β-lactam/β-lactamase inhibitors (BL/BLIs). Here are some suggestions for improvement.

1. Some sentences could be rephrased. For example, lines 38 – 41, “Among antimicrobial-resistant pathogens, carbapenem-resistant gram-negative bacteria (CR-GNB) are particularly concerning due to their resistance to almost all β-lactams and numerous other antimicrobial agents, limiting available therapeutic options”….. This could be simplified to "Carbapenem-resistant gram-negative bacteria (CR-GNB) pose a significant threat due to their resistance to multiple antimicrobial agents, limiting treatment options" to improve clarity and conciseness.

2. Line 121: The statement "Porin mutations can be accompanied by increased KPC-expression" might be misleading. Porin mutations are associated with decreased influx of antibiotics, which can contribute to resistance (DOI: https://doi.org/10.1128/cmr.00043-12). While increased expression of KPC may occur in response to antibiotic pressure, it is not directly related to porin mutations.

3. Line 122: The statement "The presence of multi-drug efflux pumps in the absence of porin downregulation does not result in the development of resistance" could be clarified. While efflux pumps alone may not lead to resistance, they can generally contribute to decreased susceptibility when combined with other resistance mechanisms.

4. Line 136 should be corrected to specify that the combination of ceftazidime/avibactam (CZA) with aztreonam (AZT) is used to treat infections caused by MBL-producing organisms, rather than infections caused by MBLs themselves.

5. Line 141: The statement "although avibactam does not inhibit MBL action, in vitro studies demonstrate that exposure of MBL bacteria to avibactam increases bacterial clearance by innate immunity mechanisms" could be further supported by specific references to these studies.

6. Line 152: The statement "However, caution is warranted when using carbapenems for such isolates since in vitro data suggest that exposure of such isolates to these antimicrobials can select for carbapenem-resistant subpopulations which retain their CZA resistance" could be further supported by specific references to these in vitro data.

7. Line 175: The statement "Ceftolozane is susceptible to the action of all carbapenemases" may be misleading. Ceftolozane is not directly affected by carbapenemases, but rather it remains effective against strains that produce carbapenemases. It is not "susceptible" to carbapenemases in the same way that other antibiotics may be susceptible to enzymatic degradation.

8. Lines 244-246: The statement about difficulties in detection and treatment of double carbapenemase producers could be expanded to explain why these strains are challenging to manage clinically.

9. Line 271-279: The statement about the differences between screening MIC and zone diameter cutoffs and clinical breakpoints could be elaborated to explain why certain carbapenemase producers can be classified as susceptible based on clinical cutoff criteria despite harboring resistance genes.

Reviewer 2 Report

Comments and Suggestions for Authors

The manuscript entitled "Antimicrobial and diagnostic stewardship of the novel β-lactam/β-lactamase inhibitors for infections due to carbapenem-resistant Enterobacterales species and Pseudomonas aeruginosa" presents an overview of the indications and current applications of the new antimicrobial combinations and explores diagnostic limitations regarding both carbapenem resistance detection and interpretation of MIC results. Moreover, the use of alternative, narrower spectrum antibiotics based on susceptibility testing is suggested and the data regarding the effect of synergies between BL/BLIs and other antimicrobials are presented. Finally, to address the absence of a standardized approach for using the novel BL/BLIs, a diagnostic and therapeutic algorithm was suggested, which can be modified based on local epidemiological criteria. 

The manuscript is comprehensive and written well. The minor suggestions are:

- Please provide full name of MIC in the abstract (line 20).

- Please provide the full name of bacteria when mentioned first and abbreviate the genus name for the rest. 

Reviewer 3 Report

Comments and Suggestions for Authors

Stefanos Ferous and colleagues (antibiotics-2916573) provide a high quality review of the antimicrobial and diagnostic stewardship of the novel β-lactam/β-lactamase inhibitors for clinically critical infections, with a particular extension and exemplifying in carbapenem-resistant Enterobacterales species and Pseudomonas aeruginosa. The overall study is interesting to read, and the suggestions and a proposal for a diagnostic and therapeutic algorithm are valuable add-ups in the field. I have a few minor comments:

1. I would suggest including carbapenem-Salmonella or E.coli in the proposed analysis, 

2. "algorithm" is a more bioinformatic or mathematical word. Please use  framework instead.

3. High-quality figures, with an organized and condensed style, are needed. The figure is wordy; please improve with a readable cartoon or real picture, the detailed figure legends can be used to explain the idea.

4. check for all word abbreviations for their first appearance, eg MICs, and more.

5. the references could be reformatted. 

Reviewer 4 Report

Comments and Suggestions for Authors

This reviewer enjoyed reading this manuscript reviewing the indications and current applications of the new antimicrobial combinations and explore diagnostic limitations regarding both carbapenem resistance detection and interpretation of MIC results. In total, 120 references were cited, and the author provided also meaningful tables and figures.

The only comment from this reviewer is that the manuscript title emphasize the importance of Pseudomonas aeruginosa in this topic. Correlating to this effort, the authors shall consider emphasizing this pathogen (Pseudomonas aeruginosa) among other bacterial strains in addressing this topic.
